# Human Fcγ-receptor IIb modulates pathogen-specific versus self-reactive antibody responses in lyme arthritis

Heike Danzer[1†], Joachim Glaesner[2†], Anne Baerenwaldt[3], Carmen Reitinger[1], Anja Lux[1], Lukas Heger[4], Diana Dudziak[4,5], Thomas Harrer[6], André Gessner[2], Falk Nimmerjahn[1,5]*

[1]Institute of Genetics, Department of Biology, University of Erlangen-Nuremberg, Erlangen, Germany; [2]Institute of Medical Microbiology and Hygiene, University Regensburg, Regensburg, Germany; [3]Laboratory for Cancer Immunotherapy, University Hospital Basel, Basel, Switzerland; [4]Department of Dermatology, Laboratory of Dendritic Cell Biology, University Hospital Erlangen, Erlangen, Germany; [5]Medical Immunology Campus Erlangen, University of Erlangen-Nuremberg, Erlangen, Germany; [6]Medical Department 3, University Hospital Erlangen, Erlangen, Germany

*For correspondence:
falk.nimmerjahn@fau.de

[†]These authors contributed equally to this work

Competing interests: The authors declare that no competing interests exist.

**Abstract** Pathogen-specific antibody responses need to be tightly regulated to generate protective but limit self-reactive immune responses. While loss of humoral tolerance has been associated with microbial infections, the pathways involved in balancing protective versus autoreactive antibody responses in humans are incompletely understood. Studies in classical mouse model systems have provided evidence that balancing of immune responses through inhibitory receptors is an important quality control checkpoint. Genetic differences between inbred mouse models and the outbred human population and allelic receptor variants not present in mice; however, argue for caution when directly translating these findings to the human system. By studying *Borrelia burgdorferi* infection in humanized mice reconstituted with human hematopoietic stem cells from donors homozygous for a functional or a non-functional FcγRIIb allele, we show that the human inhibitory FcγRIIb is a critical checkpoint balancing protective and autoreactive immune responses, linking infection with induction of autoimmunity in the human immune system.

## Introduction

B cells are an essential part of the adaptive immune system and antibody responses mounted during microbial infections protect the host from re-infection, ideally inducing sterile immunity. Loss of humoral tolerance is associated with the development of several severe autoimmune diseases, including rheumatoid arthritis and systemic lupus erythematosus (*Ludwig et al., 2017*). A variety of well-defined inbred mouse model systems have firmly established that autoantibodies can directly cause tissue inflammation and destruction (*Hogarth and Pietersz, 2012*; *Nimmerjahn and Ravetch, 2008*; *Takai, 2002*). Of note, there is a link between infections and the induction of autoimmune diseases, suggesting that during the initiation of pathogen-specific immune responses self-reactive responses may be triggered (*Brownlie et al., 2008*; *Smith and Clatworthy, 2010*; *Waisberg et al., 2011*). One prime example for such a scenario is an infection with *Borrelia burgdorferi*, which can lead to the development of joint inflammation and induction of autoimmune arthritis (*Arvikar et al., 2017*; *Steere and Glickstein, 2004*). Thus, checkpoints must be in place to limit the production of self-reactive immune responses while simultaneously promoting pathogen-specific immune system activation. Studies in inbred mouse model systems have identified the inhibitory Fcγ-receptor IIb

(FcγRIIb) expressed on B cells as one key factor for regulating humoral immune responses at several stages of B cell development (*Daëron et al., 1995*; *Smith and Clatworthy, 2010*; *Takai, 2002*; *Tarasenko et al., 2007*). Mice with altered FcγRIIb expression or a B-cell-specific/ubiquitous deletion of FcγRIIb show higher levels of IgM and IgG antibody responses, have an impaired germinal center reaction, and develop a systemic lupus erythematosus (SLE) like disease on susceptible genetic backgrounds (*Bolland and Ravetch, 2000*; *Bolland et al., 2002*; *Espéli et al., 2012*; *Li et al., 2014*; *Su et al., 2004a*; *Su et al., 2004b*; *Takai et al., 1996*). In contrast, mice with a general or a B-cell-specific overexpression of FcγRIIb showed reduced autoantibody production and autoimmune disease development (*Brownlie et al., 2008*; *McGaha et al., 2005*). As mice generated on the C57BL/6 background developed a less pronounced autoimmune phenotype it was suggested that other genes act in concert with FcγRIIb to break humoral tolerance in mice (*Boross et al., 2011*). Apart from regulating the level and quality of antibody responses, isolated triggering of FcγRIIb on plasma cells was shown to induce apoptosis, suggesting that FcγRIIb may also regulate plasma cell homeostasis (*Xiang et al., 2007*).

In humans, the level of inhibitory Fcγ-receptor IIb expression correlated with the development and severity of several autoimmune diseases. Patients with SLE and chronic inflammatory demyelinating polyneuropathy, for example, were shown to express lower levels of FcγRIIb on mature B cells and failed to upregulate FcγRIIb expression on memory B cells (*Mackay et al., 2006*; *Tackenberg et al., 2009*). Moreover, polymorphisms in the fcgr2b promoter and in the FcγRIIb transmembrane domain have been associated with the development of SLE (*Floto et al., 2005*; *Kono et al., 2005*; *Niederer et al., 2010*; *Su et al., 2007*). Interestingly, the human FcγRIIb-232T allele, in which an isoleucine residue in the transmembrane domain of the receptor is replaced with a threonine residue, resulting in an altered plasma membrane localization and decreased receptor function, was shown to be enriched in areas with malaria (*Clatworthy et al., 2007*; *Waisberg et al., 2011*; *Willcocks et al., 2010*). Thus, under negative selection pressure, a decreased level of FcγRIIb expression may provide a competitive advantage for mounting faster parasite-specific immune responses at the potential risk of inducing autoimmunity.

While the results obtained in the human system support a model in which human FcγRIIb may be involved in controlling both, protective and autoreactive humoral immune responses, the lack of model systems allowing to study the human immune system in vivo has prevented more in depth analysis (*Lux and Nimmerjahn, 2013*). To study this in more detail in the context of a human immune system, we made use of a humanized mouse model system, in which animals are reconstituted with hematopoietic stem cells (HSC) from donors carrying either the fully functional (FcγRIIb-232I) or the functionally impaired allelic variant (FcγRIIb-232T) (*Baerenwaldt et al., 2011*). By infecting humanized mice harbouring functional or non-functional FcγRIIb allelic variants with *Borrelia burgdorferi (B. burgdorferi)*, we now show that mice with a non-functional FcγRIIb allele mount greater T-cell-independent pathogen-specific antibody responses leading to a lower pathogen burden. Of note, humanized mice with impaired FcγRIIb function developed strong autoreactive antibody responses during infection, suggesting that human FcγRIIb is regulating both, the quality and quantity of human humoral immune responses. Extending these observations to the human clinical situation, we further demonstrate that humans infected with *Borrelia spec.* also developed an autoantibody response in parallel to the initiation of pathogen-specific antibody responses.

## Results

### The human immune system ameliorates lyme arthritis in humanized mice

To study human FcγRIIb function in vivo, we chose a humanized mouse model of Lyme borreliosis. In humans and select mouse strains, such as severe combined immunodeficient (SCID) mice, an infection with *Borrelia burgdorferi* (*B. burgdorferi*) triggers an inflammatory response (Lyme arthritis) caused by spirochete lipoproteins (*Barthold et al., 1990*; *Schaible et al., 1989*; *Steere and Glickstein, 2004*). If a treatment with antibiotics early after infection is not initiated or fails, this represents a major health issue. Based on previous studies in inbred mouse model systems, it has been suggested that the humoral immune system and especially the early T-cell-independent IgM and IgG3 response plays a critical role in controlling *B. burgdorferi* spread (*Barthold et al., 1996*;

*Barthold et al., 2006*; *Fikrig et al., 1997*; *LaRocca and Benach, 2008*; *McKisic and Barthold, 2000*). Furthermore, non-obese diabetic (NOD)/SCID/γc-/- (NSG) mice transplanted with a human immune system were shown to develop a relapsing fever phenotype upon infection with *Borrelia hermsii* similar to the human disease (*Vuyyuru et al., 2011*). These findings suggest that hematopoietic stem cell (HSC) humanized mice may provide a suitable model system to study whether human FcγRIIb controls pathogen and concomitant self-reactive immune responses during an infection with B. *burgdorferi*. To test if humanized NSG mice are able to respond to an infection with *B. burgdorferi*, humanized and non-humanized NSG mice were infected subcutaneously via needle injection in the right hind foot pad and followed for signs of Lyme disease, which is characterized by an initial unilateral arthritis in the ankle tissue of the infected side followed by systemic pathogen spread through the blood resulting in inflammation of the contralateral joint. As shown in *Figure 1A and B* both, non-humanized and humanized mice developed progressive inflammation starting in the infected (right foot) ankle joint followed by swelling of the contralateral non-infected (left foot) joint. If mice were humanized, however, joint inflammation of both paws was strongly delayed, suggesting that the human immune system participates in limiting pathogen burden. While one week after infection of humanized mice *B. burgdorferi* was still largely confined to the infected joint (and in about half of the animals detectable in the blood), two weeks after infection bacterial spread to the blood, heart and ears became detectable. Around 5 weeks after infection, *B. burgdorferi* was very prominent in skin (ears) and in the left foot (*Figure 1C,D*), consistent with the initiation of inflammation in the contralateral joint (*Figure 1B*). Concomitant with the infection, humanized mice developed a human IgM response directed against a variety of *B. burgdorferi* antigens including p39 and the outer surface protein C (OspC), which was comparable to the IgM response detectable in *B. burgdorferi* infected patients (*Figure 1E*). Furthermore, human and mouse immune cell infiltrates, consisting of mouse neutrophils and human myeloid cells, B cells, and CD4+ and CD8+ T cells could be detected in the joints of infected mice (*Figure 2—figure supplements 1* and *2*). Compared to the blood, especially T cells and B cells showed an activated phenotype, identified by increased expression of CD69 (*Figure 2B,C,H,I,K,L*). In contrast, no major change in serum complement C3 levels was observed during the course of infection (*Figure 2—figure supplement 2B*). In summary, these results suggest that cells of the human innate and adaptive immune system respond to the infection with *B. burgdorferi* and may help in limiting pathogen burden in humanized mice in vivo.

## The humoral immune response is critical for controlling *Borrelia burgdorferi* burden

Next, we assessed if the *B. burgdorferi*-specific antibody response participates in limiting pathogen burden. To generate humanized mice selectively lacking human B cells, we depleted B cells with the CD20-specific antibody Rituximab starting 2 days before and during the first week of infection (*Figure 3A*; *Lux et al., 2014*). While this 1-week cycle of B cell depletion resulted in a long-term reduction of B cells in humanized mice without infection, in infected animals peripheral blood B cell counts started to increase after stopping Rituximab injection and reached pre-depletion levels within 3 weeks (*Figure 3B*). In contrast to the rapid recovery of peripheral blood B cells, the reduction of serum IgM and impaired production of *B. burgdorferi*-specific antibodies was maintained over 4 weeks and only started to recover around 5 weeks post B cell depletion therapy (*Figure 3C,D*). Of note, we also detected the initiation of autoreactive B cell responses against glucose 6-phosphate isomerase (GPI) and double-stranded DNA during the course of *B. burgdorferi* infection (*Figure 3E, F*). In parallel to the suppression of the *B. burgdorferi*-specific antibody response by Rituximab treatment, autoreactive antibody responses were diminished. With respect to pathogen control, the transient depletion of B cells resulted in a higher level of pathogen positive animals. Thus, 2 weeks after infection 60% of B cell depleted animals tested positive for *B. burgdorferi* in ear tissue, compared to only 30% of animals with a functional B cell compartment (*Figure 3—figure supplement 1A,B*). Moreover, the pathogen load in ears and in the heart was reduced compared to non B-cell depleted animals until the end of the experiment (*Figure 3—figure supplement 1C,D*). With respect to joint inflammation, the transient reduction in the humoral immune response resulted in a mild increase of joint swelling especially of the non-infected joint 4 to 5 weeks after infection (*Figure 3G*).

Antibody responses may be either triggered via T-cell-dependent or T-cell-independent pathways. Indeed, FcγRIIb is also expressed on human dendritic cells (DC) present in the spleen of humanized mice (*Figure 3—figure supplement 2A*). Thus, an impaired FcγRIIb activity on DCs may

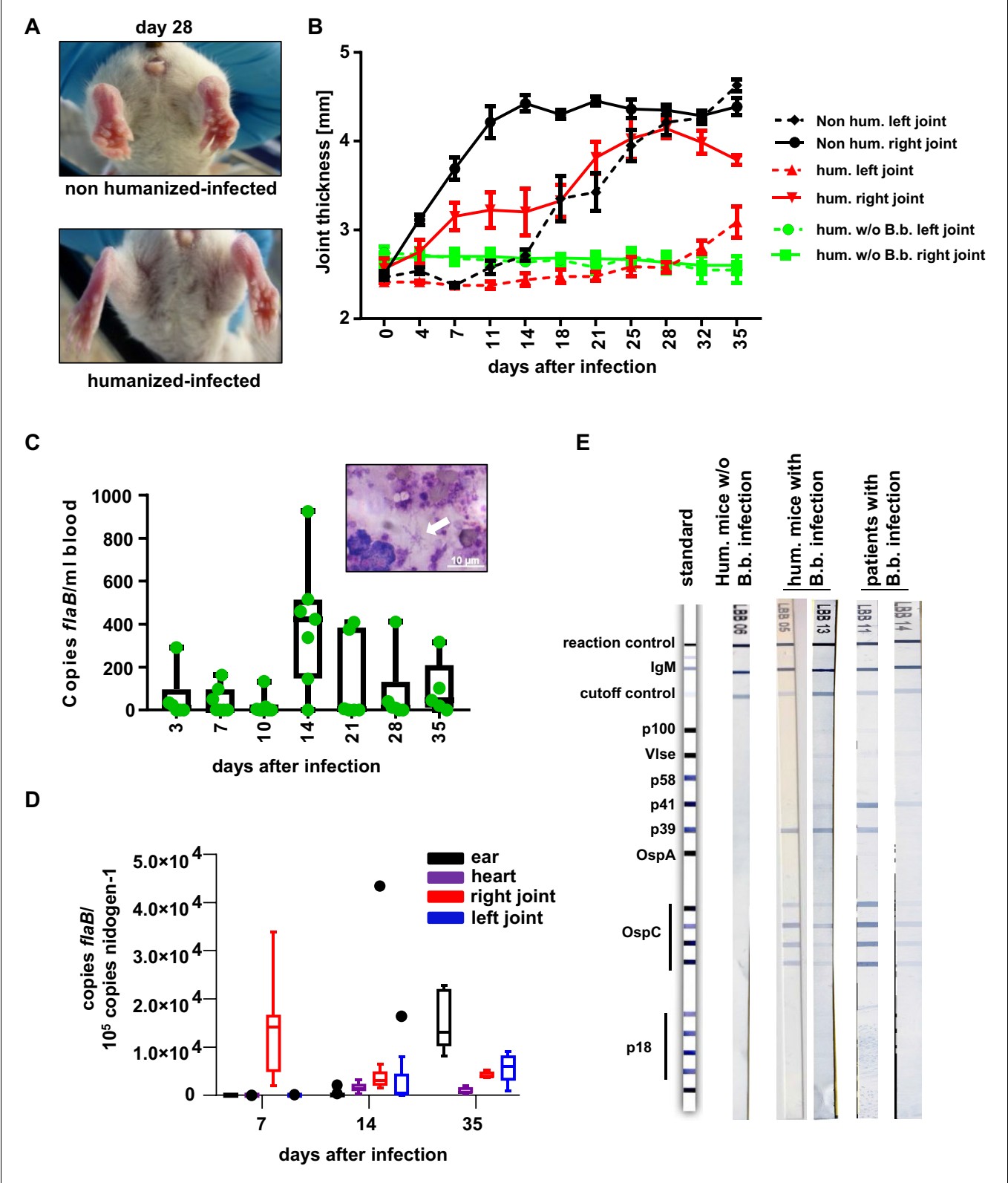

**Figure 1.** The human immune system controls *B. burgdorferi* infection. (**A, B**) Humanized and non-humanized mice were infected with *B. burgdorferi* and followed for signs of joint inflammation and pathogen spread. In (**A**) representative pictures of the hind limbs of non-humanized and humanized mice 28 days after infection are shown. (**B**) Time course of joint swelling (shown as joint thickness in mm) of the directly infected right (solid lines) and the left ankle joints of non-infected (w/o B.b.) and infected humanized (hum.) and non-humanized (non hum.) mice. Shown is the mean +/- SEM of 6–8

*Figure 1 continued on next page*

*Figure 1 continued*

mice per group. Depicted is one representative out of three independent experiments. (C) Quantification of the pathogen load (*flaB* copy numbers per ml blood) by quantitative PCR in humanized mouse blood at the indicated time points after infection. The graph shows box and whisker plots, with whiskers indicating the minimum and maximum data points. The line in the box indicates the median, the upper and lower border of the box indicates the 25th and 75th percentile of the data points. In addition, the individual data points are depicted. Inset picture shows *B. burgdorferi* spirochetes (marked with a white arrow) in a blood smear; the white scale bar represents 10 μm. (D) Quantification of *B. burgdorferi* load in ears, heart, and joints at the indicated time points after infection of humanized mice by quantitative PCR (copies *flaB*/$10^5$ copies nidogen-1). The data is depicted as box and whisker plots (Tukey), with whiskers indicating the minimum and maximum data points. The line in the box indicates the median, the upper and lower border of the box indicates the 25th and 75th percentile of the data points. (E) Shown are representative western blots (detecting IgM) with immobilized *B. burgdorferi* antigens incubated with serum from humanized mice 35 days post-infection (hum. mice with B.b. infection) or of humanized mice not infected (hum. mice w/o B.b.) with *B. burgdorferi*. As a comparison serum samples from human patients with confirmed *B. burgdorferi* infection are depicted.

also result in a more efficient T helper cell priming and a more pronounced antibody response. To determine if CD4+ T cells were critical for the *B. burgdorferi*-specific antibody response in humanized mice, we depleted this T cell subset with a CD4-specific antibody during the entire experiment. While this resulted in a strong reduction of CD4 T cells throughout the experiment, it had no impact on the production of pathogen-specific IgM antibodies, suggesting that mostly T cell independent antibody responses were critical for controlling *B. burgdorferi* infection in humanized mice (*Figure 3—figure supplement 2B,C*). In addition to and in agreement with a previous study, CD4+ T cell depletion neither had an impact on progression of arthritis (not shown) (*Lasky et al., 2016*) nor on pathogen load (*Figure 3—figure supplement 1A,B*). Altogether, this set of experiments suggests that the early T-cell-independent humoral immune system plays an important role in limiting *B. burgdorferi* replication and pathogen induced pathology in humanized mice.

## Human FcγRIIb regulates the quality and quantity of the pathogen-specific human humoral immune response

After having established that the human humoral immune response is involved in controlling early *B. burgdorferi* infection, we next assessed if the inhibitory FcγRIIb plays a role in controlling the quality and quantity of the humoral immune response. To study this, we generated humanized mouse colonies homozygous either for the functional (FcγRIIb-232I) or the non-functional FcγRIIb allele (FcγRIIb-232T), by reconstituting NSG mice with human hematopoietic stem cells (HSC) from donors homozygous for the respective FcγRIIb alleles. Four months after HSC transplantation, both mouse colonies were infected with *B. burgdorferi* and analyzed in parallel with respect to the pathogen-induced immune response and the development of joint inflammation. As demonstrated in *Figure 4A and B*, humanized mice with the non-functional FcγRIIb allele showed a clearly detectable expansion of CD138+ plasma blasts in the blood, whereas plasma blasts did not increase in mice with the functional FcγRIIb allele. Moreover, plasma blast expansion in humanized mice with the non-functional FcγRIIb allele occurred in two separate waves peaking at 1 and 4 weeks after infection (*Figure 4B*). In line with the increased level of plasma blasts in FcγRIIb-232T humanized mice, a stronger increase in serum IgM and *B. burgdorferi*-specific IgM responses was detectable in these animals (*Figure 4C, D*). In line with the stronger humoral immune response, pathogen load in ears was reduced in FcγRIIb-232T humanized mice 2 weeks after infection (*Figure 3—figure supplement 1A,B*) and this reduced pathogen burden was maintained until the end of the experiment (day 35 after infection) (*Figure 4E,F*). However, compared to the functional FcγRIIb allele, a stronger development of self-reactive antibodies directed against double stranded DNA and GPI could be observed in humanized mice with the non-functional FcγRIIb-232T allele (*Figure 4G,H*). In summary, these results suggest that FcγRIIb function is critical to limit the production of autoreactive antibodies during an immune response against *B. burgdorferi*.

## Correlation between the induction of pathogen-specific and autoreactive immune responses in humanized mice and humans

To determine if the induction of protective and autoreactive immune responses over time coincided in individual mice carrying the functional (232I) or non-functional (232T) FcγRIIb alleles, we plotted

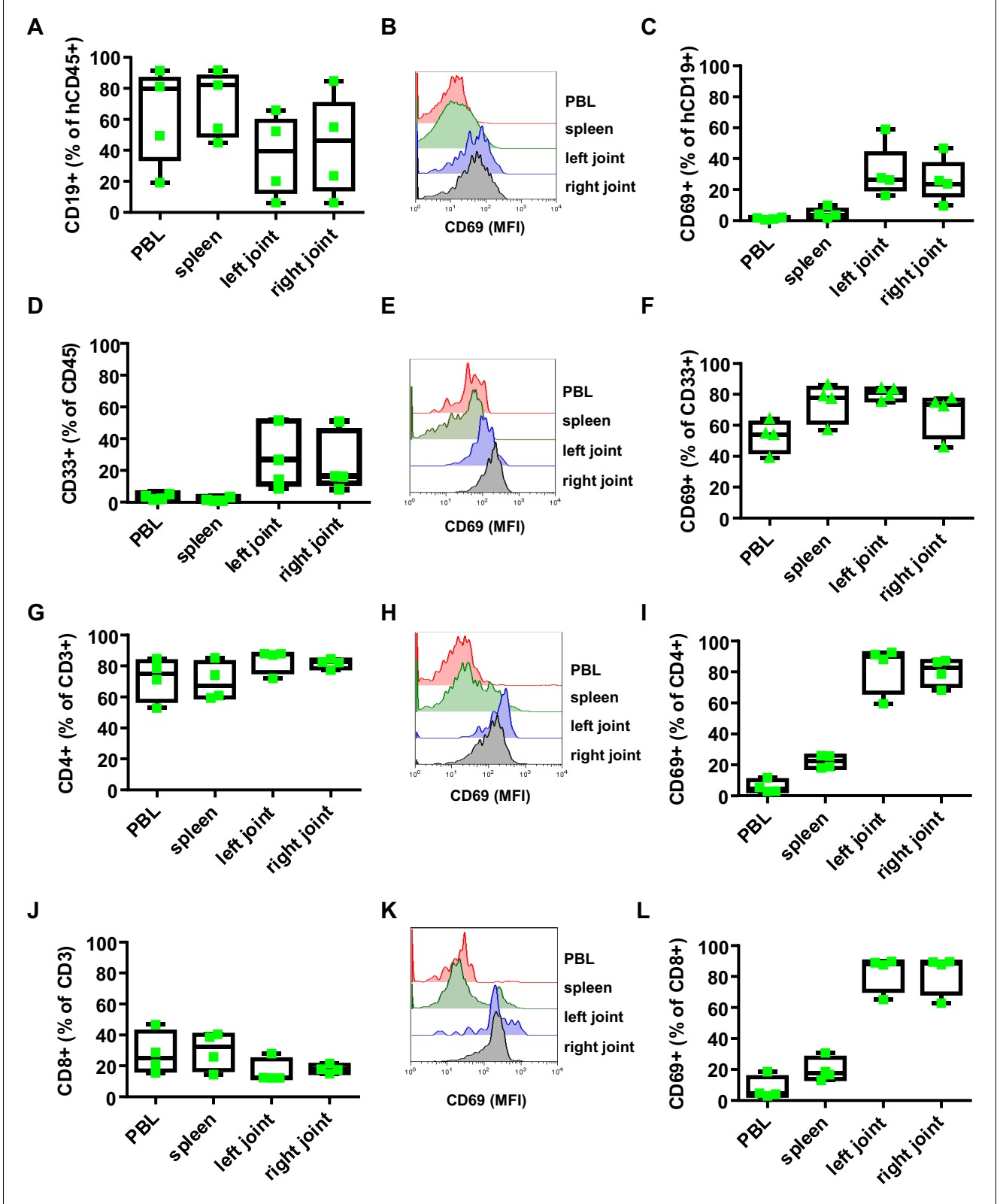

**Figure 2.** Effect of *B. burgdorferi* infection on the human immune system in humanized mice. Shown is the effect of *B. burgdorferi* infection on the abundance (percentage of all human CD45+ cells) and activation status (CD69 expression) of human CD19+ B cells (**A-C**), CD33+ myeloid cells (**D-F**), CD4+ T cells (**G-I**), and CD8+ T cells (**J-L**) in the peripheral blood (PBL), spleen, and left and right joints 38 days after infection. Representative FACS plots showing the mean fluorescence intensity (MFI) of CD69 expression on B cells (**B**), myeloid cells (**E**), CD4 T cells (**H**) and CD8 T cells (**K**) are

*Figure 2 continued on next page*

*Figure 2 continued*

presented. Depicted are box and whisker plots including all data points. The line in the box indicates the median, the upper and lower border of the box indicates the 25th and 75th percentile of the data points. Whiskers extend to minimum and maximum data points. One representative out of three independent experiments with four mice per group is shown.

The online version of this article includes the following figure supplement(s) for figure 2:

**Figure supplement 1.** Gating strategy to identify mouse and human immune cells in humanized mice.

**Figure supplement 2.** Infiltration of mouse immune cells in *B. burgdorferi* infected humanized mice.

the concentration of serum IgM against IgM reactivity against *B. burgdorferi,* double-stranded DNA and glucose 6-phosphate isomerase (*Figure 5*). This analysis confirmed that the induction of increased serum IgM levels in humanized mice upon *B. burgdorferi* infection coincided with the appearance of pathogen specific but also self-reactive immune responses (*Figures 5A,B* and *4G, H*). When further plotting *B. burgdorferi*-specific immune responses of individual mice against DNA and GPI-specific autoantibody responses a clear correlation between the induction of *B. burgdorferi* and GPI-specific antibody responses became evident across all time points and levels of reactivity, while only those animals producing the highest levels of pathogen-specific antibodies also produced antibodies directed against DNA (*Figure 5C,D*).

To test if a similar correlation between pathogen specific and autoreactive B cell responses can be detected in humans infected with *B. burgdorferi*, we compared a cohort of patients with confirmed *B. burgdorferi* infections with a group of non-infected controls (*Figure 6*). In striking similarity to the results in humanized mice, *B. burgdorferi*-specific IgM responses showed a strong correlation with the induction of GPI-specific antibody responses in the patient but not in the healthy control cohort (*Figure 6A,B*). In contrast, the correlation with the DNA-specific immune response was much less pronounced, similar to the observations in humanized mice. Unfortunately (but not unexpectedly), all of the patients in our cohort were homozygous for the functional FcγRIIb-232I allele, not allowing to study the impact of a loss of FcγRIIb function during *B. burgdorferi* infection in humans. Nonetheless, our results suggest that there is a clear correlation between the induction of pathogen-specific and autoreactive antibody responses, which is regulated by the human inhibitory FcγRIIb at least in humanized mice.

## Discussion

The aim of this study was to investigate if and how the human inhibitory FcγRIIb impacts human B cell responses during an infection with *B. burgdorferi*. While results from inbred mouse model systems strongly suggest that mouse FcγRIIb is a critical regulator of humoral and innate immune responses, they have also emphasized that the mouse genetic background can modulate the observed phenotypes (*Bolland and Ravetch, 2000*; *Bolland et al., 2002*; *Boross et al., 2011*; *Nimmerjahn and Ravetch, 2010*; *Smith and Clatworthy, 2010*). Moreover, differences in the level of FcγRIIb expression between mice and humans, with humans having a much lower level of expression especially on innate immune cells, argue for caution when transferring results from classical mouse models to the outbred human immune system (*Lux and Nimmerjahn, 2013*). To account for these interspecies differences, we have developed a humanized mouse model system in which immunodeficient mice are reconstituted with a human immune system homozygous for the functional or non-functional FcγRIIb allele (*Baerenwaldt et al., 2011*). In the present study, we used this humanized mouse model system to address how FcγRIIb modulates humoral immune responses during an episode of a bacterial infection. We used a model of *B. burgdorferi* infection, as the spread of the pathogen and the pathogen induced pathology in immunodeficient mouse strains mimics many features of the human disease (*Steere and Glickstein, 2004*). Consistent with previous studies using *Borrelia hermsii*, we demonstrate that NSG mice develop an inflammation of the infected joint, followed by systemic pathogen spread and a delayed arthritis of distant joints during later stages of infection with *B. burgdorferi* (*Barthold et al., 1990*; *Vuyyuru et al., 2011*). Humanized NSG mice showed a delayed joint inflammation suggesting that the human immune system in humanized mice participates in controlling the infection and concomitant immune pathology. Indeed, human B cells,

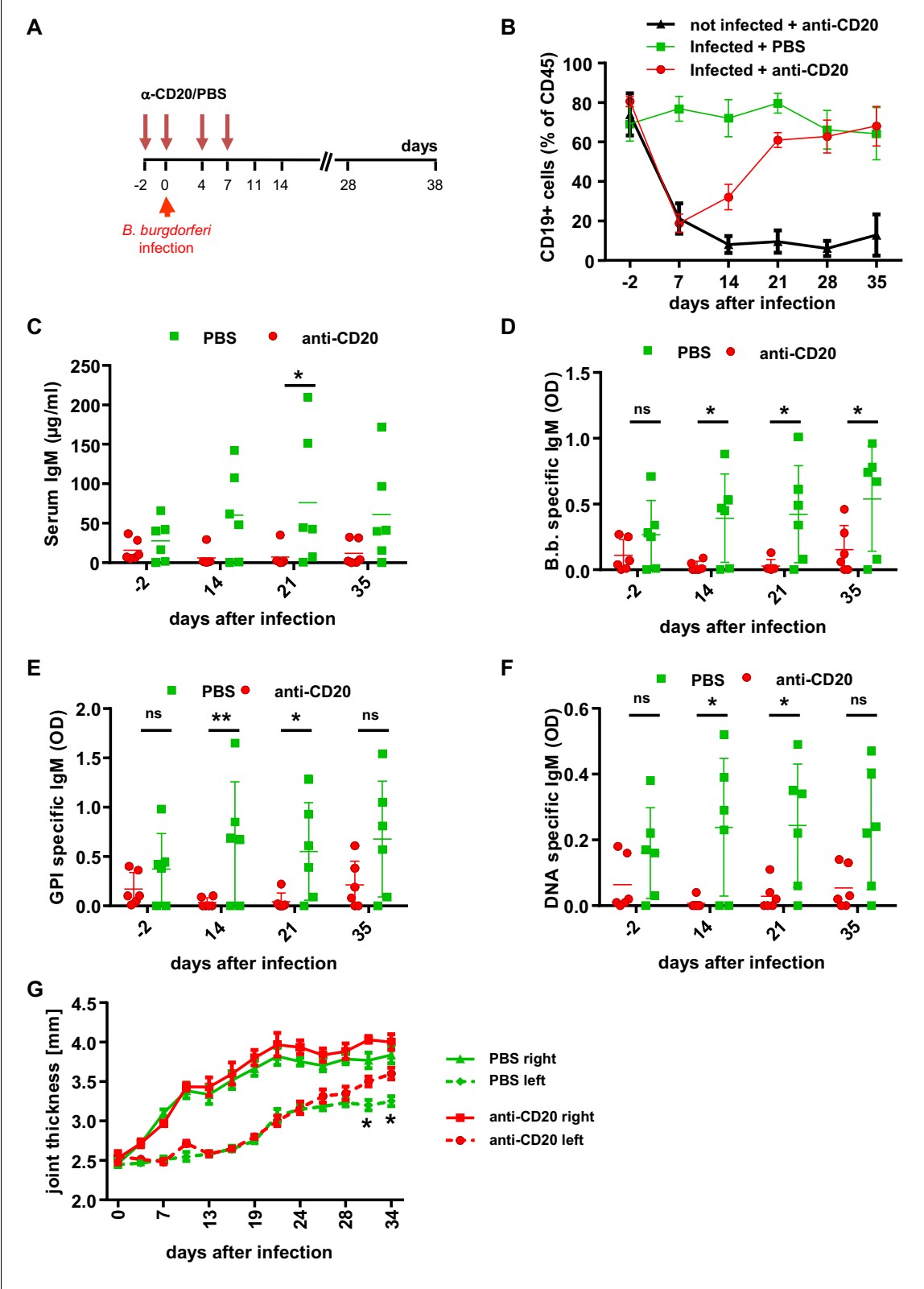

**Figure 3.** The role of B cells in *B. burgdorferi* control. (A) Schematic representation of the experimental strategy. B cells were depleted with a human CD20-specific antibody (α-CD20) injected 2 days before (d-2), at the day of infection (d0) and 4 and 7 days after infection with *B. burgdorferi*. PBS injection served as a control. (B) Shown is the percentage (mean +/- SEM) of human B cells (CD19+) among human leukocytes (CD45+) in the peripheral blood of humanized mice injected with PBS or a CD20-specific antibody in the absence (not infected) or presence (infected) of an infection
*Figure 3 continued on next page*

*Figure 3 continued*

with *B. burgdorferi*. Shown is one representative out of two independent experiments with six mice per group. (C–F) Depicted are the serum IgM levels (C), the *B. burgdorferi*- (D), glucose 6-phosphate isomerase (GPI)- (E), and double-stranded DNA (dsDNA)-specific (F) IgM responses (mean+/-SEM) in infected humanized mice treated with PBS or a CD20-specific antibody as determined by ELISA. Shown is one out of two independent experiments with n = 6 mice per group. A two-way Anova with Tukey's or Sidak's multiple comparison test was used for statistical evaluation. *p<0.05, **p<0.01, ns = not significant. (G) Shown is the joint thickness (mean+/-SEM) of the right and left hind ankle joint of humanized mice infected with *B. burgdorferi* and treated with PBS or a CD20-specific antibody (anti-CD20) during the first 7 days. A two-way Anova with Tukey's multiple comparison test was used for statistical evaluation. n = 6–10 mice per group. *p<0.05. Shown is one representative out of three independent experiments.

The online version of this article includes the following figure supplement(s) for figure 3:

**Figure supplement 1.** Effect of human FcγRIIb alleles, T cells and B cells on *B. burgdorferi* load and on pathogen-specific immune responses.
**Figure supplement 2.** FcγRIIb expression on dendritic cells and effect of human T cells on *B. burgdorferi* specific immune responses.

T cells and myeloid cells infiltrated the infected joints and showed a highly activated phenotype, similar to observations in human patients with Lyme arthritis (*Gross et al., 1998*; *Yssel et al., 1991*). As removal of B cells but not CD4+T cells impaired pathogen control, the human T helper cell independent humoral immune response was dominantly involved in limiting *B. burgdorferi* pathology, in line with previous results in inbred mouse strains (*Barthold et al., 1996*; *Barthold et al., 2006*; *LaRocca and Benach, 2008*). Altogether this supports the notion that the humanized mouse model of *B. burgdorferi* infection recapitulates important aspects of the human infection, allowing to investigate how the human inhibitory FcγRIIb controls the humoral immune response during infection in vivo. By comparing the immune response in humanized mice with functional (FcγRIIb-232I) and non-functional (FcγRIIb-232T) FcγRIIb alleles side by side, our results suggest that human FcγRIIb indeed is a critical regulator of the magnitude and quality of the human immune response in humanized mice. Thus, in the presence of the non-functional FcγRIIb-232T allele, a lower pathogen burden in the directly infected joint and in the skin, heart and distal joint was observed. Most strikingly, humanized mice with the non-functional FcγRIIb allele demonstrated a stronger and biphasic expansion of plasma blasts in the blood and correlated with a higher level *B. burgdorferi* specific antibodies in the serum. The increased level of plasma blasts in FcγRIIb-232T humanized mice may be explained via two possible mechanisms. Firstly, a broader panel of naïve B cells may become activated and develop into plasma blasts in the absence of FcγRIIb function, as has been observed in FcγRIIb-deficient mice (*Nakamura et al., 2003*; *Takai et al., 1996*). This may also explain the simultaneous priming of autoreactive B cell responses, again in line with data from FcγRIIb-deficient mice (*Bolland and Ravetch, 2000*). Interestingly, the increased appearance of autoantibodies in FcγRIIb-232T humanized mice coincided with the second wave of plasma blasts, whereas increased levels of pathogen-specific immune responses were detectable much earlier. These results may indicate that either the continued presence of the pathogen increases the likelihood to develop secondary autoreactive immune responses if FcγRIIb function is missing; or, that the induction of autoantibody responses is developing in parallel but at a reduced strength. It is important to note that in this scenario lack of FcγRIIb function on human DCs present in humanized mice may also contribute to the induction of autoreactive immune responses, by lowering the threshold for DC activation and (auto)antigen presentation (*Desai et al., 2007*; *Dhodapkar et al., 2005*). Furthermore, FcγRIIb on DCs was shown to participate in presenting intact antigen to B cells, which may be another pathway along which DCs contribute to the induction of humoral immune responses in humanized mice (*Bergtold et al., 2005*). This will need to be investigated in more detail in future studies. Secondly, FcγRIIb cross-linking on plasma cells residing in the bone marrow was shown to trigger plasma cell apoptosis, allowing newly formed plasma cells to access survival niches in the bone marrow (*Xiang et al., 2007*). Thus, in the absence of FcγRIIb function more plasma blasts/cells may accumulate in the periphery due to a reduced level of plasma cell apoptosis in the bone marrow.

Overall, our results are consistent with reports demonstrating that in humans the non-functional FcγRIIb allele is enriched in regions with malaria and protects from certain cerebral forms of the disease, while simultaneously increasing the risk to develop autoimmune diseases (*Clatworthy et al., 2007*; *Waisberg et al., 2011*; *Willcocks et al., 2010*). Indeed, especially the GPI-specific immune response showed a clear correlation with the induction of *B. burgdorferi*-specific antibodies, while the DNA-specific autoantibody responses mostly correlated in mice with a very high antibody

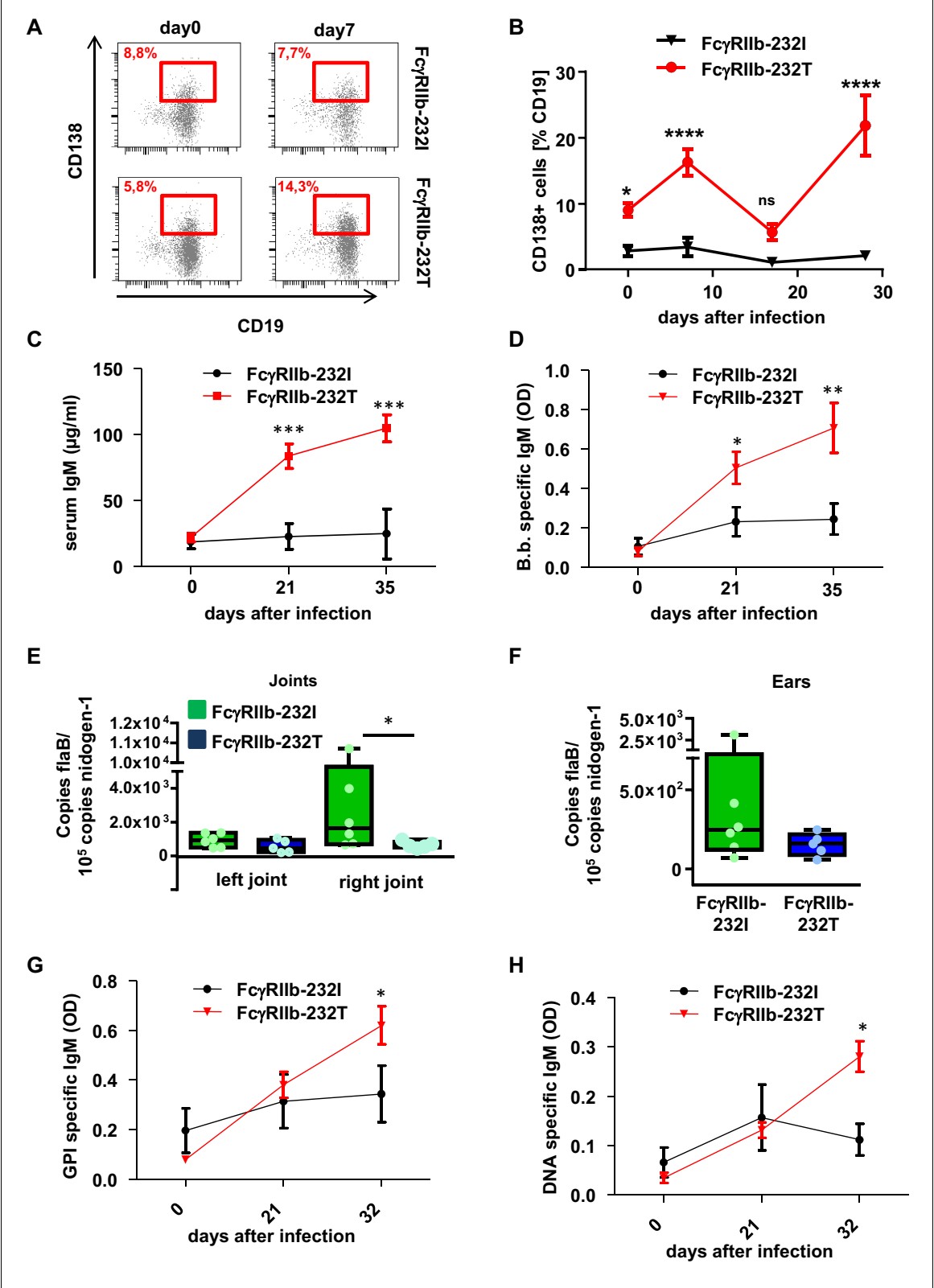

**Figure 4.** Impact of human FcγRIIb alleles on the humoral immune response and pathogen load during *B. burgdorferi* infection. (A) Shown are representative dot plots of CD19+CD138+ plasma blasts (marked by red squares including the respective percentage of CD19+CD138+ cells within the population of CD19+ B cells) in the peripheral blood of humanized mice homozygous for the non-functional (FcγRIIb-232T) or functional (FcγRIIb-232I) FcγRIIb allele before and 7 days after infection with *B. burgdorferi* as detected by flow cytometry. (B) Quantification of the abundance of CD19+CD138
*Figure 4 continued on next page*

Figure 4 continued

+ plasma blasts in the peripheral blood of humanized mice homozygous for the non-functional (FcγRIIb-232T) or functional (FcγRIIb-232I) FcγRIIb allele by FACS analysis at the indicated time points after infection. Shown are mean values +/- SEM of one representative out of two independent experiments with n = 7–12 mice per group. For statistical evaluation, a two-way Anova with a Bonferroni's multiple comparisons test was used. *p<0.05, ****p<0.0001; ns indicates no significant difference. (C) Quantification of human IgM levels in sera of humanized mice homozygous for functional (FcγRIIb-232I) and non-functional (FcγRIIb-232T) FcγRIIb alleles at the indicated timepoints after *B. burgdorferi* infection. Shown are mean values +/- SEM of one representative out of two independent experiments with n = 7–12 mice per group. For statistical evaluation, a two-way Anova with a Bonferroni's multiple comparisons test was used. ***p<0.0005; (D) Shown is the *B. burgdorferi* specific IgM response (OD) at the indicated time points after infection of humanized mice homozygous for functional (FcγRIIb-232I) and non-functional (FcγRIIb-232T) FcγRIIb alleles. Shown are mean values +/- SEM of one representative out of two independent experiments with n = 7–12 mice per group. For statistical evaluation, a two-way Anova with a Bonferroni's multiple comparisons test was used. *p<0.05; **p<0.005. (E, F) Quantification of the pathogen load (*B. burgdorferi flaB* copy numbers normalized to mouse nidogen-1 copy numbers) in ankle joints (left joint, L; right joint, (R) (E) and ears (F) of infected humanized mice homozygous for the functional FcγRIIb-232I (232I) or non-functional FcγRIIb-232T (232T) allele by quantitative PCR 35 days after infection with *B. burgdorferi*. n = 5–6 mice per group. A two-way Anova test was used to evaluate statistical significance. *p<0.05. (G, H) Detection of glucose 6-phosphate isomerase (GPI) (G) and DNA-specific (H) human IgM responses during the course of *B. burgdorferi* infection in humanized mice homozygous for the functional (FcγRIIb-232I) or non-functional (FcγRIIb-232T) FcγRIIb alleles. Shown are mean values +/- SEM of one representative out of two independent experiments with n = 7–12 mice per group. For statistical evaluation a two way Anova with a Bonferroni's multiple comparisons test was used. *p<0.05.

response. As GPI-specific immune responses have been demonstrated to be causative for inflammatory arthritis in mouse models and have been detected at least in subsets of human arthritis patients (*Korganow et al., 1999*; *Monach et al., 2004*), our findings may provide evidence for a direct link between the initial *B. burgdorferi* infection and the development of autoimmune arthritis occuring several years after the initial infection (*Arvikar et al., 2017*). While our results show that humanized mice reconstituted with hematopoietic stem cells from donors homozygous for select FcγRIIb alleles may be a valuable pre-clinical model system to study the impact of FcγRIIb function on human disease, some caveats of this model system need to be considered. First of all, apart from the difference in FcγRIIb function, the genetic background of the individual HSC donors is different. Thus, it cannot be excluded that other genes impacting pathogen recognition, humoral tolerance or immune cell activation may contribute to the observed phenotypes. This, however, will be similar in a human patient population and could only be circumvented if transgenic mice selectively expressing the human FcγRIIb alleles on a mouse FcγRIIb knockout background would be used. A second major difference to a human patient population is that the human immune system in humanized mice has been developing only for three months. Thus, not all immune cell subsets have reached their steady-state abundance, which is mirrored by the fact that B cells and IgM antibody responses dominate the immune response during this early developmental stage. Especially the low level of IgG antibody responses may result in a more moderate disease pathology as only IgG antibodies are efficiently able to recruit FcγR-dependent effector pathways. Furthermore, humanized mice are kept under well-controlled conditions avoiding infections, which human patients will have encountered during their lifetime and which may alter consecutive immune responses. Despite all these points to be considered, the fact that the humoral immune response observed in humanized mice closely resembles at least in part the antibody response in humans, underscores the value of this pre-clinical model system.

In summary, our study revealed that the human inhibitory FcγRIIb is critical to limit self-reactive immune responses during the early phase of an infection with *B. burgdorferi*. Considering that a link between a primary infection with *B. burgdorferi* and the development of arthritis later in life has been suggested it will be of great interest to study if humans with this non-functional FcγRIIb allele are more prone to develop arthritis, while they are more resistant to infection at the same time. As the presence of this polymorphism in the Caucasian population is rare, the challenge will be to obtain enough samples from *B. burgdorferi* infected patients homozygous for the non-functional FcγRIIb allele. We are currently increasing the number of new and longitudinal serum samples in our patient cohort to address if humans with the non-functional FcγRIIb232T allele are more prone to develop autoreactive antibody responses during the primary infection with *B. burgdorferi* and Lyme arthritis later in life.

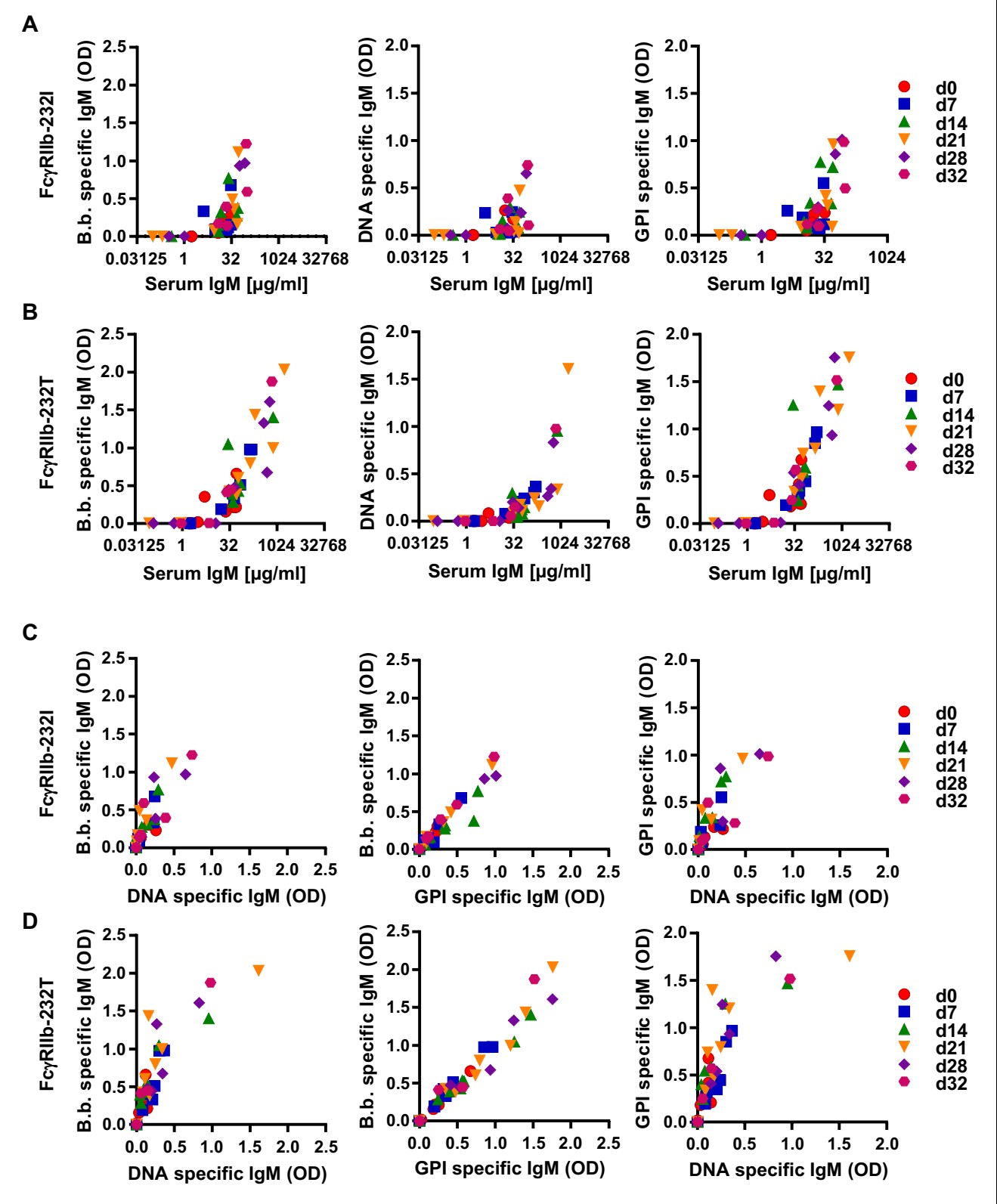

**Figure 5.** Correlation of human autoreactive and pathogen-specific antibody responses in humanized mice. Humanized mice homozygous for the functional (FcγRIIb-232I; n = 9) (A) and non-functional (FcγRIIb-232T; n = 12) (B) FcγRIIb alleles were infected with *B. burgdorferi* and serum samples were collected before (d0) and 7, 14, 21, 28, and 32 days after infection. Data points from the same timepoint are depicted with the same symbol as indicated in the Figure legend at the right side of the graph. (A, B) Depicted is the correlation between total serum IgM (µg/ml) with *B. burgdorferi*

*Figure 5 continued on next page*

*Figure 5 continued*

specific, DNA specific and GPI specific IgM responses (OD) in individual humanized mice homozygous for the functional (FcγRIIb-232I) (**A**) and non-functional (FcγRIIb-232T) (**B**) FcγRIIb alleles at different time-points of infection as determined by ELISA. (**C, D**) Shown is the correlation between the induction of *B. burgdorferi* specific IgM responses with concomitant DNA- or GPI-specific IgM responses in individual humanized mice homozygous for the functional (FcγRIIb-232I) (**C**) or non-functional (FcγRIIb-232T) (**D**) FcγRIIb alleles at different time-points after infection.

# Materials and methods

## Key resources table

| Reagent type (species) or resource | Designation | Source or reference | Identifiers | Additional information |
|---|---|---|---|---|
| Antibody | anti-human CD3 PE/Cy7 (mouse monoclonal) | BioLegend | Cat.#:300316 | FACS (1:2000) |
| Antibody | anti-human CD4 APC (rat monoclonal) | BioLegend | Cat.#:357408 | FACS (1:200) |
| Antibody | anti-human CD8a FITC (mouse monoclonal) | BioLegend | Cat.#:301050 | FACS (1:100) |
| Antibody | Anti-human CD11c FITC (mouse monoclonal) | BioLegend | Cat.#:301604 | FACS (1:50) |
| Antibody | Anti-human FcγRIIb PE (recombinant human IgG1-N297A) | in house | Clone 2B6 | FACS (1:50) |
| Antibody | anti-human CD19 Brilliant Violet 510 (mouse monoclonal) | BioLegend | Cat.#:302242 | FACS (1:100) |
| Antibody | anti-human CD20 FITC (mouse monoclonal) | BioLegend | Cat.#:302304 | FACS (1:200) |
| Antibody | anti-human CD33 PE (mouse monoclonal) | BioLegend | Cat.#:303404 | FACS (1:200) |
| Antibody | anti-human CD33 BV510 (mouse monoclonal) | BioLegend | Cat.#:303422 | FACS (1:100) |
| Antibody | anti-human CD45 APC/Fire750 (mouse monoclonal) | BioLegend | Cat.#:304062 | FACS (1:400) |
| Antibody | anti-human CD69 PerCP/Cy5.5 (mouse monoclonal) | BioLegend | Cat.#:310926 | FACS (1:100) |
| Antibody | anti-human CD138 PE/Cy7 (mouse monoclonal) | BioLegend | Cat.#:356514 | FACS (1:200) |
| Antibody | anti-mouse/human CD11b PerCP/Cy5.5 (rat monoclonal) | BioLegend | Cat.#:101228 | FACS (1:1000) |
| Antibody | anti-mouse CD45.1 PE (mouse monoclonal) | BioLegend | Cat.#:110707 | FACS (1:600) |
| Antibody | anti-mouse CD45.1 Brilliant Violet 421 (mouse monoclonal) | BioLegend | Cat.#:110732 | FACS (1:400) |
| Antibody | anti-mouse CD62L PE/Cy7 (rat monoclonal) | BioLegend | Cat.#:104418 | FACS (1:1000) |
| Antibody | anti-mouse F4/80 APC/Fire 750 (rat monoclonal) | BioLegend | Cat.#:123152 | FACS (1:200) |
| Antibody | anti-mouse Gr-1 Brilliant Violet 510 (rat monoclonal) | BioLegend | Cat.#:108437 | FACS (1:300) |
| Antibody | anti-mouse Ly-6G FITC (rat monoclonal) | BioLegend | Cat.#:127606 | FACS (1:300) |

*Continued on next page*

*Continued*

| Reagent type (species) or resource | Designation | Source or reference | Identifiers | Additional information |
|---|---|---|---|---|
| Antibody | 9E9 Alexa647 (rabbit monoclonal) | *Nimmerjahn et al., 2005* | n/a | FACS (1:200) |
| Antibody | Anti-human CD4 (for depletion) (mouse monoclonal) | BioXCell | Cat.#:BE0003-2 | 100 µg per injection |
| Antibody | Anti-human CD20 (for depletion) (recombinant mouse monoclonal) | *Kao et al., 2015* | In house | 25 µg per injection |
| Strain, strain background *Mus musculus* | NOD.Cg-Prkdc<scid > Il2rg<sup>tm1Wjl/Szj</sup> | The Jackson Laboratory | Cat.#:005557 | n/a |
| Software, algorithm | Graph Pad Prism 7.03 | GraphPad Software Inc, San Diego, CA, USA | | |

## Experimental model and subject details

### Mice

NOD.Cg-Prkdc<scid > Il2rg$^{tm1Wjl/Szj}$ (NSG) mice (The Jackson Laboratories) were held under specific pathogen-free (SPF) conditions in isolated ventilated cages (IVC) in the animal facility of the Friedrich Alexander University Erlangen-Nuremberg according to the rules and regulations of the German animal welfare law. All animal experiments were approved by the government of lower Franconia.

### Human hematopoietic cord blood

For humanization of NSG mice, human hematopoietic stem cells (HSC) were purified from cord blood. Cord blood samples were provided by the Women's Hospital Fürth, Germany, and the Women's hospital of the University Hospital Erlangen. Cord blood samples were collected anonymously with informed consent of the donor and the local ethical committee.

### Human serum

Human serum samples from patients with *B. burgdorferi* infection or from control individuals were collected with approved consent and were provided in an anonymous fashion by the Medical Department 3 of the University Hospital Erlangen (Rheumatology and Immunology, Prof. Dr. Thomas Harrer). While sera of 17 patients with *B. burgdorferi* infection were collected, the number of samples from healthy individuals was restricted to eleven samples due to ethical considerations.

## Method details

### Generation of humanized mice

Humanized mice were generated as described (*Baerenwaldt et al., 2011*). In brief, CD34+ HSCs were isolated from cord blood with the 'Direct CD34 Progenitor Cell Isolation Kit, human' (Miltenyi Biotec) according to the manufacturers instruction. Newborn NSG mice were irradiated with 1.4 Gy and 20,000–50,000 CD34+ HSCs were injected into the facial vein. Mice were analyzed after 12–16 weeks for the presence of human cells using an anti-human CD45 antibody (Biolegend, see key resource table). For the infection with *B. burgdorferi*, mice with more than 5% human cells containing B cells, T cells and myeloid cells were randomly allocated into experimental groups.

### Isolation of genomic DNA and genotyping

Genotyping of cord blood samples for FcγRIIb alleles was done as described before (*Baerenwaldt et al., 2011*). In brief, genomic DNA was isolated with the „QiaAmp DSP Blood Mini Kit "(Qiagen, Hilden) following the instructions of distributor. FcγRIIB genotyping was carried out with a two-step PCR protocol. Briefly, a 15 kb product was amplified using the Qiagen 'LongRange PCR Kit' (Qiagen) (Primers LongRange fwd: ctccacaggttactcgtttctaccttatcttac and LongRange rev: gcttgcgtggccctggttctca) generating a 14.7 kb amplicon which was purified via gel electrophoresis

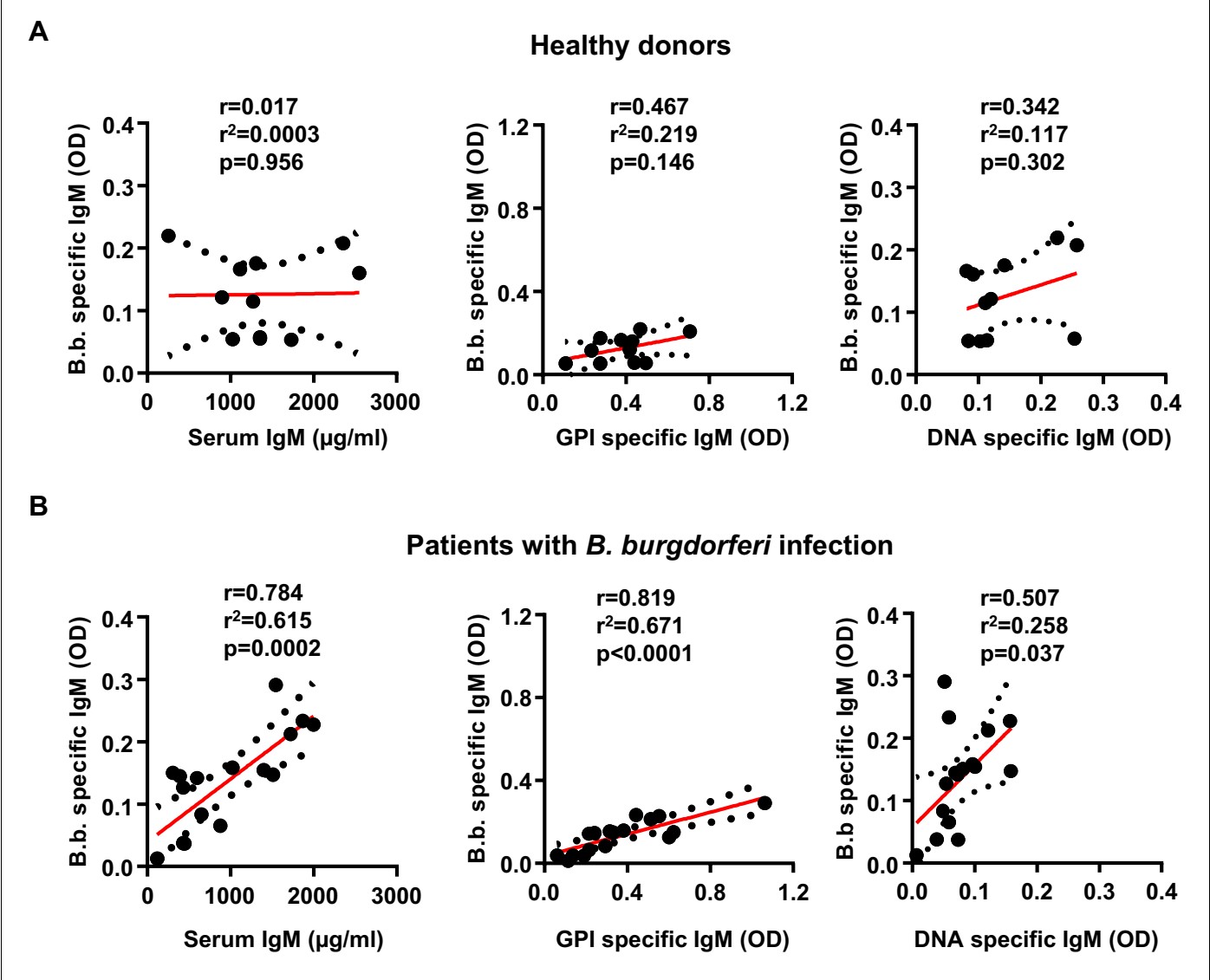

**Figure 6.** Induction of *B. burgdorferi*-specific and autoreactive antibody responses in humans. Shown is the correlation of serum IgM (µg/ml), glucose 6-phosphate (OD), and DNA specific (OD) IgM responses with *B. burgdorferi* specific IgM responses in healthy donors (n = 11) (A) and a cohort of patients with confirmed *B. burgdorferi* (n = 17) (B) infections. Graphs include all datapoints and the linear regression curve (red line) and error bar lines (depicting the SEM as black dotted lines). After testing for normal distribution of the individual data sets with a D'Agostino and Pearson normality test, the statistical evaluation for a correlation between the two parameters was performed with a Pearson's correlation test. Pearson's r, $r^2$ and p values are indicated within each graph.

and by using the Qiagen 'Gel purification Kit'. This PCR product was used as a template for the nested PCR to amplify the transmembrane region (Primers R2Btm-fwd: aaggggagcccttccctctgtt and R2Btm-rev: gtggcccctggttctcagaa). Subsequently, the PCR product (2365 bp) was gel purified and sequenced (using the R2Bseq primer: aaggggagcccttccctctgtt).

### Infection with *B. burgdorferi*

The N40 isolate of *B. burgdorferi* was grown at 30°C in BSK-H medium containing 6% rabbit serum (Bio and Sell) and underwent fewer than five in vitro passages before inoculation. Spirochetes were enumerated with a cell counting chamber by phase contrast microscopy and diluted with sterile medium. Mice were inoculated subcutaneously with $5 \times 10^5$ bacteria in 50 µl of BSK-H into the right hind footpad.

## Measurement of joint swelling

The development of joint swelling as a correlate for arthritis, was monitored in a blinded fashion by measuring the thickness of both (infected and contralateral) tibiotarsal joints using a metric caliper (Kroeplin). Measurements were taken in the anterior-posterior orientation with extended ankles through the thickest part of the joint.

## Determination of spirochete load in organs by quantitative PCR

DNA was extracted from organs of infected mice using the QIAamp DNA Mini kit (Qiagen) according to the manufacturer's instructions with the following modifications regarding the initial processing of individual samples:

Blood: 150 µl of blood was taken from the retroorbital plexus of mice on days 7, 14, 21, 28, 32, and 40 after inoculation and deposited into heparinized tubes. Thereafter, samples were centrifuged at 500 x g for 5 min to pellet blood cells, whereas spirochetes were remaining in the plasma. After transfer to a new tube, the plasma was centrifuged at 5000 x g for 10 min to pellet bacteria. The supernatant was discarded, and the pellet was suspended in 200 µl PBS. Bacterial cells were lysed at 56°C for 10 min by adding 20 µl of proteinase K and 200 µl of AL buffer provided with the kit.

Ears: On day 14 after inoculation, approximately one-third of an earlobe was removed and digested o/n at 56°C in 380 µl ATL buffer and 20 µl proteinase K provided in the kit. The remaining ear tissue was removed at final sacrifice and treated in the same way.

Hearts: Hearts were removed without prior organ perfusion. After cutting along the longitudinal axis, one half of the heart was used for DNA extraction. Tissue was digested o/n at 56°C by adding 380 µl ATL/20 µl proteinase K directly followed by bead-beating in a TissueLyser II (Qiagen) for 2 min at 30 Hz.

Joints: Rear hind limbs were excised just above and below the tibiotarsal joint. After removal of hair and bone and thorough dissection with scissors, pieces of tissue (representing mainly the superficial part of the joint tissue) were digested o/n at 56°C in 380 µl ATL/20 µl proteinase K. DNA was eluted from the column membranes with 50 µl (blood samples) and 100 µl (all other samples), respectively, of AE buffer (10 mM Tris-HCl, 0.5 mM EDTA; pH 9.0).

Quantification of *B. burgdorferi* transcripts was conducted with the 7900HT Real-Time PCR System (Applied Biosystems by Thermo Fisher Scientific) using the following primers: forward, 5'-TCTTTTCTCTGGTGAGGGAGCT-3' and reverse, 5'-TCCTTCCTGTTGAACACCCTCT-3', which amplify a 70 bp fragment of the *Borrelia flagellin B* gene (*flaB*, chromosomal, single copy); and forward, 5'-CCAGCCACAGAATACCATCC-3' and reverse, 5'-GGACATACTCTGCTGCCATC-3', which flank a product 154 bp in length corresponding to the murine *nidogen-1* (*nid-1*) gene that was used as reference for normalization. HPLC-purified primers were provided by Thermo Fisher Scientific. PCRs for *flaB* and *nid-1* were performed synchronously using the LightCycler 480 SYBR Green I Master kit (Roche). The 10 µl reaction volume contained 5 µl ready-to-use master mix (2 x concentrated), 0.4 µM of each primer, 2 µl of $H_2O$, and 2 µl of extracted DNA in a dilution specified in the respective figure legends or 2 µl of external standard template. DNA preparations extracted from the reference strain of *B. burgdorferi* sensu stricto were used to establish a standard curve for the *flaB* PCR. Serial dilutions of a pGEM-T Easy plasmid (Promega) harboring a mouse *nid-1* fragment were included as standards in each normalization PCR for *nid-1*. Applying the SDS 2.4.1 software (Applied Biosystems), the number of spirochetes was calculated on the basis of the standards and normalized to $10^5$ copies of *nid-1*.

The amplification program consisted of the initial denaturation step at 95°C for 5 min and 40 cycles of denaturation at 95°C for 10 s, annealing at 60°C for 15 s and extension at 72°C for15 s. Fluorescence was measured at the end of each extension step. After each amplification, melting curves were acquired to determine the specificity of PCR products.

## Detection of spirochetes in organs by conventional nested PCR

DNA was extracted from blood and ears as described for the real time PCR approach above. The *flagellin B* gene of *Borrelia burgdorferi* N40 was detected in organs by nested PCR followed by analysis of amplicons on an ethidium-bromide-stained agarose gel. 50 ng of purified DNA was used as a template in each reaction for the first PCR using the primers Flagellin-P1 (5'-CTGCTGGCATGGGAGTTTCT-3') and Flagellin-P2 (5'-TCAATTGCATACTCAGTACT-3'). The amplification program

consisted of the initial denaturation step at 94°C for 3 min and 35 cycles of denaturation at 94°C for 30 s, annealing at 51°C for 30 s, extension at 72°C for 60 s, and a final extension at 72°C for 10 min producing a 730 bp fragment. 3 µl of PCR product from the first reaction were used as template in a second PCR using primers Flagellin-P3 (5'-AAGGAATTGGCAGTTCAATC-3') and Flagellin-P4 (5'-ACAGCAATAGCTTCATCTTG-3'). The amplification program consisted of the initial denaturation step at 94°C for 3 min and 35 cycles of denaturation at 94°C for 30 s, annealing at 58°C for 30 s, extension at 72°C for 30 s, and a final extension at 72°C for 10 min producing a 290 bp fragment. The reactions were performed using Platinum *Taq* DNA Polymerase PCR master mix (Invitrogen by Thermo Fisher Scientific) according to the manufacturer's instructions. Band sizes were determined by a GeneRuler 100 bp DNA ladder (Thermo Fisher Scientific). A *Borrelia burgdorferi*-positive tick lysate was used as a positive control.

## B cell depletion in humanized NSG mice

To deplete B cells, 25 µg of the CD20 specific antibody Rituximab was injected at day −2, 0, 4 and 7 post infection. To analyze the efficacy of B cell depletion, peripheral blood was diluted 1:2 with PBS and PBMCs were isolated using density gradient centrifugation with Pancoll (PAN Biotech). PBMCs were stained with anti-human CD19 (BD) and anti-human CD3 (BD) antibodies and samples were analyzed with a FACS Canto II (BD). Analysis was performed using FlowJo software (Tristar).

## CD4+ T cell depletion in humanized NSG mice

To deplete T cells, 100 µg of the human CD4-specific antibody OKT4 (BioXcell) was injected intraperitoneally 3, 2 and 1 days before *B. burgdorferi* infection and at 10, 17, 24 and 32 days after infection. The success of CD4 T cell depletion was monitored in the peripheral blood by FACS analysis.

## Quantification of antibody responses by ELISA

Sera of humanized mice were obtained by collecting blood from the retro-orbital plexus. Serum from control and *B. burgdorferi* infected humans was obtained by collecting venous blood in BD Vacutainer tubes at the Medical Department 3 of the University Hospital Erlangen. Serum samples were generally stored at −80°C until further use. For quantification of total serum IgM in humanized mice and humans the Bethyl 'Human IgM ELISA Quantitation Kit' (Biomol) were used according to the instructions of the manufacturer. OD was measured with 'VersaMax tunable microplate reader' (Molecular Devices) at 450 and 650 nm.

## Detection of *B. burgdorferi*-specific antibodies

For detection of *B. burgdorferi*-specific antibodies, ELISA plates were coated with *B. burgdorferi* ultrasonic lysate at 5 µg/ml (0.5 µg/well) over night at 4°C. After washing, plates were blocked with PBS/3%BSA for 2 hr at room temperature, followed by removal of blocking solution and addition of sera from humanized mice or humans (1:100 dilution in PBS) for 30 min at room temperature. After washing three times with PBS, bound IgM was detected using the HRP coupled IgM-specific antibody from the Bethyl human IgM quantification kit or the human IgG-specific antibody from the Biomol human IgG quantification kit (both at a 1:10000 dilution in PBS/3%BSA). Antibodies against a predefined series of immunodominant proteins of *B. burgdorferi* sensu *lato* were measured using a commercial western blot approach in which purified recombinant *B. burgdorferi* antigens are coated on a membrane which is incubated with the sera of humanized mice (diluted 1:20), followed by detection of bound antibodies by human IgM-HRP coupled secondary antibodies.

## Detection of dsDNA specific antibodies by ELISA

For the detection of anti-DNA antibodies, ELISA plates were coated with 10 µg/ml methylated BSA (Sigma) in PBS for 2 hr at room temperature. After washing, the plates were coated with 50 µg/ml calf thymus DNA (Sigma) in PBS at 4°C overnight. Blocking of unspecific binding was performed with PBS/0.1% Gelatine/3% BSA/1 mM EDTA for 2 hr at RT. Sera were diluted 1:100 in the blocking solution and incubated for 1 hr at room temperature. After three washing steps with PBS, an HRP-conjugated IgM-specific antibody (from the Bethyl human IgM Quantitation Kit) was used at a dilution of 1:10,000 (in blocking solution) for detection. After an incubation for 1 hr at room temperature, plates

were washed three times with PBS and bound antibody was detected with TMB Solution (from the Bethyl human IgM Quantitation Kit). The reaction was stopped with 6% orthophosphoric acid.

## Detection of GPI-specific antibodies by ELISA

Elisa plates were coated with 100 µl of a 5 µg/ml glucose 6-phosphate isomerase (GPI) stock solution in PBS at 4°C over night. After three washing steps with PBS, blocking was performed with 100 µl of PBS/3% BSA for 1 hr at room temperature. After removal of the blocking solution, sera were added at a 1:100 dilution in PBS/3% BSA and incubated for 1 hr at room temperature. After three washing steps with PBS, the detection antibody (from the Bethyl human IgM Quantitation Kit) was used at a 1:10,000 dilution in PBS/3% BSA for 1 hr at RT. Detection of bound human antibodies was performed as described for the anti-DNA ELISA.

## Detection of mouse complement C3

Elisa plates were coated with 100 µl of a 1:1000 dilution of goat anti-mouse complement C3 (Cappel laboratories, distributed by MP Biomedicals) in Carbonate-Bicarbonate Buffer (Sigma) at 4°C overnight. After three washing steps with PBS, blocking was performed with 200 µl of PBS/3% BSA for 1 hr at room temperature. After removal of the blocking solution, sera were added at a 1:300 dilution in PBS/1% BSA and incubated for 1 hr at room temperature. After three washing steps with PBS, the HRP conjugated detection antibody (goat anti-mouse complement C3 from Cappel laboratories) was used at a 1:10,000 dilution in PBS/1% BSA for 1 hr at RT. Detection of bound human antibodies was performed as described for the anti-DNA ELISA.

## Detection of *B. burgdorferi*-specific humoral immune responses via western blot

Antibodies against a series of conserved *B. burgdorferi* proteins were measured using a commercial western blot approach (RecomLine Borrelia IgM kit, Mikrogen, Neuried, Germany) according to the instructions of the manufacturer. Sera of humanized mice and humans we used at a dilution of 1:20 in the supplied sample dilution buffer and bound antibodies were detected by using the provided human IgM-HRP coupled secondary antibody.

## Enzymatic isolation of joint and spleen cells for FACS analysis

After excision of joints and complete removal of skin and muscle tissue, joints were cut into small pieces and incubated for 2 hr in 1.5 ml Collagenase D (1 mg/ml in HBSS) at room temperature. Every 30 min, the incubation was mixed thoroughly by vortexing. After the incubation 2 ml of RPMI 1640 medium was added and the cell suspension was filtered through a 70 µM cell strainer, followed by pelleting the cells at 1400 rpm for 5 min at room temperature. For subsequent FACS analysis, the cell pellet was re-suspended in 150 µl FACS buffer (PBS/2% FCS, 0.02% sodium azide).

## Flow-cytometric analysis of human leukocyte populations

Flow cytometric measurements were performed on single-cell suspensions of blood, spleen and joint cells using a FACS Canto II (BD Biosciences, San Jose, CA). Fifteen minutes prior to staining with fluorochrome-coupled antibodies, the cells were incubated on ice for 10 min with Fc-block (anti-FcRγIII/FcγRIV, clone 2.4G2; 10 µg/ml) in FACS buffer (PBS/2% FCS, 0.02% sodium azide) to prevent unspecific binding to Fc receptors. All antibody stainings were carried out at 4°C in FACS buffer. Data acquisition and analysis was performed with the FACS Diva software (BD Biosciences, San Jose, CA, USA). A complete list of antibodies can be found in the key resources table. Representative gating strategies for analyzing mouse and human immune cell subpopulations are shown in Figure S1.

## Statistical analysis

All data are means ± standard error of the mean (SEM). Graphs and statistical analysis were performed using GraphPad Prism 7.03 software (GraphPad Software Inc, San Diego, CA). All samples were tested for Gaussian distribution. Two-way analysis of variance (ANOVA) was used for comparison of multiple groups. If samples were not normally distributed, the Mann-Whitney-test was used for comparing two groups of samples or the Kruskal-Wallis-test was used respectively for comparing

multiple groups. For linear regression analysis, a D'Agostino and Pearson normality test was performed to test for normal distribution, followed by a Pearson's (normal data distribution) or Spearman's (non-normal data distribution) test for correlation. A detailed description of statistical tests used for individual experiments can be found in the respective Figure legends.

## Acknowledgements

We are grateful to Heike Albert for expert technical assistance. This work was funded by the German Research Foundation (TRR130-P13 to FN and FOR2886-B2 to AL and FN) and the Bavarian Ministry of Science and Art (Bayresq.Net to DD and AG).

## Additional information

### Funding

| Funder | Grant reference number | Author |
|---|---|---|
| Deutsche Forschungsgemeinschaft | TRR130-P13 | Falk Nimmerjahn |
| Deutsche Forschungsgemeinschaft | FOR 2886 | Falk Nimmerjahn |
| Deutsche Forschungsgemeinschaft | FOR2886 | Anja Lux |
| Bavarian State Ministry of Education, Science and the Arts | | Diana Dudziak André Gessner |

The funders had no role in study design, data collection and interpretation, or the decision to submit the work for publication.

### Author contributions

Heike Danzer, Data curation, Investigation, Methodology; Joachim Glaesner, Conceptualization, Resources, Data curation, Validation, Investigation, Visualization; Anne Baerenwaldt, Data curation, Investigation, Writing - original draft; Carmen Reitinger, Resources, Data curation, Validation, Investigation; Anja Lux, Resources, Data curation, Validation; Lukas Heger, André Gessner, Resources, Formal analysis, Investigation, Methodology; Diana Dudziak, Falk Nimmerjahn, Conceptualization, Resources, Formal analysis, Supervision, Funding acquisition, Investigation, Writing - original draft; Thomas Harrer, Resources, Investigation

### Author ORCIDs

André Gessner ⓘ http://orcid.org/0000-0003-4316-2408
Falk Nimmerjahn ⓘ https://orcid.org/0000-0002-5418-316X

### Ethics

Animal experimentation: All animal experiments were performed in strict accordance to the rules and regulations of the German animal welfare law. All animal experiments were approved by the government of lower Franconia (Permit Numbers: 2532-2-469 and 2532.2-817-11).

### Decision letter and Author response

Decision letter https://doi.org/10.7554/eLife.55319.sa1
Author response https://doi.org/10.7554/eLife.55319.sa2

## Additional files

### Supplementary files

• Source data 1. Summarized source data for all figures.

- Transparent reporting form

## Data availability

All data generated and analysed during the study are included in the manuscript. Source data has been provided for all figures.

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
