## [Decision Letter]

**Acceptance summary:**

Although the role of FcgRII in humoral immunity and tolerance has been well characterized in the mouse model system, simply extrapolating these data to the human system is somewhat dangerous. Therefore, in this study, authors employed humanized mouse system and tried to clarify the function of FcgRII. The data were quite convincing and the results improved our understanding for the mechanisms balancing protective and autoreactive immune responses.

**Decision letter after peer review:**

Thank you for submitting your article "Human Fcγ-receptor RIIb modulates pathogen-specific versus self-reactive antibody responses in Lyme arthritis" for consideration by *eLife*. Your article has been reviewed by three peer reviewers, and the evaluation has been overseen by a Reviewing Editors and Tadatsugu Taniguchi as the Senior Editor. The following individuals involved in review of your submission have agreed to reveal their identity: Toshiyuki Takai (Reviewer #2) and Wanli Liu (Reviewer #3).

The reviewers have discussed the reviews with one another and the Reviewing Editor has drafted this decision to help you prepare a revised submission.

Summary:

Although the role of fcgRII in humoral immunity and tolerance has been well characterized in the mouse model system, simply extrapolating these data to the human system is somewhat dangerous. Therefore, in this stduy, authors employed humanized mouse system and tried to clarify the function of FcgRII. These results improved our understanding for the mechanisms balancing protective and autoreactive immune responses. This study is a good candidate for *eLife*. However, the following points should be addressed before publication.

Essential revisions:

1) Quantification of *B. burgdorferi* load in ears, heart, and joints was only showed at the time points of day 14 and 35 in Figure 1. Since *B. burgdorferi* infection is a dynamic and complicated process in mice, a complete time course including early days after the change shall be elucidated in detail?

2) The human IgM response recognizing *B. burgdorferi* in Figure 1E is confusing. It shall be provided with the details including the measure time or the detailed time point so that the human immune system responses can be better integrated into this manuscript.

3) To validate that the humoral immune response is critical for controlling *B. burgdorferi* spread, it is required to show the *B. burgdorferi* spreading condition at the corresponding time points in Figure 3. In addition, the data in Figure 3 was acquired 38 days after infection and it is required to clarify any significantly changes before and after that time point?

---

## [Author Response]

Summary:Although the role of fcgRII in humoral immunity and tolerance has been well characterized in the mouse model system, simply extrapolating these data to the human system is somewhat dangerous. Therefore, in this stduy, authors employed humanized mouse system and tried to clarify the function of FcgRII. These results improved our understanding for the mechanisms balancing protective and autoreactive immune responses. This study is a good candidate for eLife. However, the following points should be addressed before publication.

We sincerely thank the reviewers for these encouraging statements.

Essential revisions:1) Quantification of B. burgdorferi load in ears, heart, and joints was only showed at the time points of day 14 and 35 in Figure 1. Since B. burgdorferi infection is a dynamic and complicated process in mice, a complete time course including early days after the change shall be elucidated in detail?

We thank the reviewers for this insightful and important comment. We have now performed new experiments analyzing *B. burgdorferi* spread starting from day 3 in the serum and from day 7 in the joints, heart and ear. As shown in the new Figure 1D, this now creates a better time resolved picture of how *B. burgdorferi* spreads in humanized mice, starting from the right joint at day 7, followed by a systemic spread starting at day 14.

2) The human IgM response recognizing B. burgdorferi in Figure 1E is confusing. It shall be provided with the details including the measure time or the detailed time point so that the human immune system responses can be better integrated into this manuscript.

We apologize for having this made not clear enough. We now incorporated a more detailed Figure legend stating the exact time points, when we analyzed the serum IgM responses. We further included human serum samples to show that a similar *B. burgdorferi* specific antibody response occurs in humans.

3) To validate that the humoral immune response is critical for controlling B. burgdorferi spread, it is required to show the B. burgdorferi spreading condition at the corresponding time points in Figure 3. In addition, the data in Figure 3 was acquired 38 days after infection and it is required to clarify any significantly changes before and after that time point?

We fully agree that our experiments do not allow concluding that *B. burgdorferi* spread is altered. In fact, the previous and also the newly added experiments show that the bacterial load but not the spreading is reduced. We have now included new data quantifying *B. burgdorferi* infection of heart and ears between 24 and 35 days after infection, demonstrating a clear trend towards a lowered bacterial burden, but not a change in organ tropism or spread. We have now edited the text accordingly. As our animal permit does not allow us to perform further experiments with B cell depleting antibodies we would hope that the new data and adjusted text is now sufficient for the reviewer. A new animal license would take at least a year for approval under normal circumstances and under the current Corona pandemic even longer.